# Effectiveness of mRNA boosters after homologous primary series with BNT162b2 or ChAdOx1 against symptomatic infection and severe COVID-19 in Brazil and Scotland: A test-negative design case–control study

Thiago Cerqueira-Silva[1,2‡], Syed Ahmar Shah[3‡], Chris Robertson[4,5‡], Mauro Sanchez[6], Srinivasa Vittal Katikireddi[4,7], Vinicius de Araujo Oliveira[2,8], Enny S. Paixão[9], Igor Rudan[3], Juracy Bertoldo Junior[2,8], Gerson O. Penna[6], Neil Pearce[9], Guilherme Loureiro Werneck[10,11], Mauricio L. Barreto[2,8], Viviane S. Boaventura[1,2‡], Aziz Sheikh[3‡], Manoel Barral-Netto[1,2,8‡]*

1 LIB and LEITV Laboratories, Instituto Gonçalo Moniz, Fiocruz, Salvador, Bahia, Brazil, 2 Universidade Federal de Bahia (UFBA), Salvador, Bahia, Brazil, 3 Usher Institute, University of Edinburgh, Edinburgh, United Kingdom, 4 Public Health Scotland, Glasgow, United Kingdom, 5 Department of Mathematics and Statistics, University of Strathclyde, Glasgow, United Kingdom, 6 Universidade de Brasília, Brasília, Distrito Federal, Brazil, 7 MRC/CSO Social & Public Health Sciences Unit, University of Glasgow, Glasgow, United Kingdom, 8 Center for Data Integration and Knowledge for Health (Cidacs), Instituto Gonçalo Moniz, Fiocruz, Salvador, Bahia, Brazil, 9 London School of Hygiene and Tropical Medicine, London, United Kingdom, 10 Universidade do Estado do Rio de Janeiro, Rio de Janeiro, Brazil, 11 Universidade Federal do Rio de Janeiro, Rio de Janeiro, Brazil

‡ TCS, SAS, and CR share first authorship on this work. VSB, AS and MB-N are joint senior authors on this work.
* manoel.barral@fiocruz.br

## Abstract

### Background

Brazil and Scotland have used mRNA boosters in their respective populations since September 2021, with Omicron's emergence accelerating their booster program. Despite this, both countries have reported substantial recent increases in Coronavirus Disease 2019 (COVID-19) cases. The duration of the protection conferred by the booster dose against symptomatic Omicron cases and severe outcomes is unclear.

### Methods and findings

Using a test-negative design, we analyzed national databases to estimate the vaccine effectiveness (VE) of a primary series (with ChAdOx1 or BNT162b2) plus an mRNA vaccine booster (with BNT162b2 or mRNA-1273) against symptomatic Severe Acute Respiratory Syndrome Coronavirus 2 (SARS-CoV-2) infection and severe COVID-19 outcomes (hospitalization or death) during the period of Omicron dominance in Brazil and Scotland compared to unvaccinated individuals. Additional analyses included stratification by age group (18 to 49, 50 to 64, ≥65). All individuals aged 18 years or older who reported acute respiratory illness symptoms and tested for SARS-CoV-2 infection between January 1, 2022, and

**Data Availability Statement:** Regarding Brazilian data availability, one of the study coordinators (M. B.-N.) signed a term of responsibility on using each database made available by the Ministry of Health (MoH). Each member of the research team signed a term of confidentiality before accessing the data. Data was manipulated in a secure computing environment, ensuring protection against data leakage. The Brazilian National Commission in Research Ethics approved the research protocol (CONEP approval no. 4.921.308). Our agreement with the MoH for accessing the databases patently denies authorization of access to a third party. Any information for assessing the databases must be addressed to the Brazilian MoH at https://datasus. saude.gov.br/, and requests can be addressed to datasus@saude.gov.br. In this study, we used anonymized secondary data following the Brazilian Personal Data Protection General Law, but it is vulnerable to re-identification by third parties as they contain dates of relevant health events regarding the same person. To protect the research participants' privacy, the approved Research Protocol (CONEP approval no. 4.921.308) authorises the dissemination only of aggregated data, such as the data presented here. Regarding Scotland, the data that support the findings of this study are not publicly available because they are based on de-identified national clinical records. These are, however, available by application via Scotland's National Safe Haven from Public Health Scotland. The data used in this study can be accessed by researchers through NHS Scotland's Public Benefit and Privacy Panel via its Electronic Data Research and Innovation Service.

**Funding:** The present study was suported by Fiocruz and partly supported by a donation from the "Fazer o Bem Faz Bem" programme from JBS S.A. MB-N received a grant from Fundação de Apoio do Estado da Bahia (FAPESB) – Grant PNX0008/2014/ Fapesb, Edital 08/2014 - Programa de Apoio a Núcleos de Excelência. GLW acknowledges Fundação Carlos Chagas Filho de Amparo à Pesquisa do Estado do Rio de Janeiro (E-26/210.180/2020). This study is part of the EAVE II project. EAVE II is funded by the MRC (MC_PC_19075) (AS, SVK, CR) with the support of BREATHE—The Health Data Research Hub for Respiratory Health (MC_PC_19004) (AS), which is funded through the UK Research and Innovation Industrial Strategy Challenge Fund and delivered through Health Data Research UK. This research is part of the Data and Connectivity National Core Study, led by Health Data Research UK in partnership with the Office for National Statistics and funded by UK Research and Innovation (grant ref MC_PC_20058) (AS, SVK, CR). Additional

April 23, 2022, in Brazil and Scotland were eligible for the study. At 14 to 29 days after the mRNA booster, the VE against symptomatic SARS-CoV-2 infection of ChAdOx1 plus BNT162b2 booster was 51.6%, (95% confidence interval (CI): [51.0, 52.2], $p < 0.001$) in Brazil and 67.1% (95% CI [65.5, 68.5], $p < 0.001$) in Scotland. At $\geq$4 months, protection against symptomatic infection waned to 4.2% (95% CI [0.7, 7.6], $p = 0.02$) in Brazil and 37.4% (95% CI [33.8, 40.9], $p < 0.001$) in Scotland. VE against severe outcomes in Brazil was 93.5% (95% CI [93.0, 94.0], $p < 0.001$) at 14 to 29 days post-booster, decreasing to 82.3% (95% CI [79.7, 84.7], $p < 0.001$) and 98.3% (95% CI [87.3, 99.8], $p < 0.001$) to 77.8% (95% CI [51.4, 89.9], $p < 0.001$) in Scotland for the same periods. Similar results were obtained with the primary series of BNT162b2 plus homologous booster. Potential limitations of this study were that we assumed that all cases included in the analysis were due to the Omicron variant based on the period of dominance and the limited follow-up time since the booster dose.

## Conclusions

We observed that mRNA boosters after a primary vaccination course with either mRNA or viral-vector vaccines provided modest, short-lived protection against symptomatic infection with Omicron but substantial and more sustained protection against severe COVID-19 outcomes for at least 3 months.

## Author summary

### Why was this study done?

- Brazil and Scotland have been offering boosters for the population that received two doses of vaccines against the coronavirus that causes Coronavirus Disease 2019 (COVID-19). However, after Omicron (a SARS-CoV-2 variant) emerged, both countries reported a high number of COVID-19 cases despite accelerating their booster programs.

- Knowledge about the duration of the protection offered by the booster doses is essential to guide public health recommendations.

### What did the researchers do and find?

- We analyzed national databases from Brazil and Scotland between January and April 2022 to estimate the protection offered by mRNA booster doses in individuals who received a primary series of viral vector or mRNA anti-COVID-19 vaccines.

- For individuals that received primary series of viral vector vaccine plus mRNA booster, from 14 to 29 days to $\geq$4 months after the booster dose, vaccine effectiveness (VE) against symptomatic infection decreased significantly in Brazil from 51.6%, 95% confidence interval (CI): [51.0, 52.2], to 4.2% (95% CI: [0.7, 7.6], $p = 0.02$) and in Scotland from 67.1% (95% CI [65.5, 68.5], $p < 0.001$) to 37.4% (95% CI [33.8, 40.9], $p < 0.001$).

support has been provided through Public Health Scotland and Scottish Government Director General Health and Social Care and National Core Studies - Immunology. The original EAVE project was funded by the National Institute for Health Research (NIHR) Health Technology Assessment programme (11/46/23) (AS). The Brazilian component is part of the Fiocruz VigiVac project on continuous digital evaluation of the national anti-COVID-19 immunization programme. SVK acknowledges funding from an NHS Research Scotland Senior Clinical Fellowship (SCAF/15/02), the MRC (MC_UU_00022/2), and the Scottish Government Chief Scientist Office (SPHSU17). CR reports grants from the Medical Research Council (MRC) and Public Health Scotland during the conduct of the study. ESP is funded by the Wellcome Trust [Grant number 213589/Z/18/Z]. This partnership between Brazil and Scotland was established through funding from the NIHR (GHRG /16/137/99) using UK aid from the UK Government to support global health research(AS, SVK, CR, IR). The funders had no role in study design, data collection and analysis, decision to publish, or preparation of the manuscript.

**Competing interests:** I have read the journal's policy and the authors of this manuscript have the following competing interests: VdAO, VB, MLB, and MB-N are employees of Fiocruz, a federal public institution, which manufactures Vaxzevria in Brazil, through a full technology transfer agreement with AstraZeneca. Fiocruz allocates all its manufactured products to the Ministry of Health for the public health service use. SVK was a member of the UK Government's Scientific Advisory Group on Emergencies subgroup on ethnicity, the Cabinet Office's International Best Practice Advisory Group, and was co-chair of the Scottish Government's Expert Reference Group on Ethnicity and COVID-19. CR is a member of the Scottish Government Chief Medical Officer's COVID-19 Advisory Group, Scientific Pandemic Influenza Group on Modelling, and Medicines and Healthcare products Regulatory Agency Vaccine Benefit and Risk Working Group. CR reports the followings: "Research Grants to Strathclyde University from Public Health Scotland, UK Medical Research Council, Scotland Chief Scientist Office, Health Data Research UK. Advisory Bodies: Member of UK SPI-M committee, Scottish Government Scientific Advisory Committee, MHRA Covid vaccine benefit and risk expert working group." IR is the member of the Advisory scientific committee on COVID-19 of the Government of Croatia and co-Editor-in-Chief of the Journal of Global Health. AS is an Academic Editor on PLOS Medicine's editorial board, and is a member of the

- In these periods, a slight decrease in VE was observed against severe outcomes in Brazil from 93.5% (95% CI [93.0, 94.0], $p < 0.001$) to 82.3% (95% CI [79.7, 84.7], $p < 0.001$) and in Scotland from 98.3% (95% CI [87.3, 99.8], $p < 0.001$) to 77.8% (95% CI [51.4, 89.9], $p < 0.001$). Similar results were obtained with a homologous booster after a primary series of mRNA vaccines.

- Similar findings in two very different countries allow us to draw reliable results because of potential sources of bias in effectiveness studies, such as differences in testing behavior and unmeasured characteristics between vaccinated and unvaccinated individuals, which are unrelated in the two countries.

## What do these findings mean?

- Modest, short-lived protection was observed against symptomatic infection caused by the Omicron variant after two doses of either vector viral or mRNA vaccine plus a booster dose with mRNA vaccine. However, protection against hospitalization or death was substantial for at least 3 months.

## Introduction

The effectiveness of available Coronavirus Disease 2019 (COVID-19) vaccines may differ by variants of concern (VOCs) and by waning immunity. Before the emergence of the Omicron VOC, real-world vaccine effectiveness (VE) studies had reported substantial protection against symptomatic infection and severe outcomes (i.e., hospitalization and death) [1–3]. However, the protection offered by COVID-19 vaccines has been shown to wane over time [4,5], prompting many countries to provide booster doses [6]. With Omicron's emergence and rapid spread, the booster program was expedited and expanded in several countries. A few studies have evaluated the protection offered after the booster, although with conflicting results [7–9].

The available, still limited, body of evidence indicates a rapid waning of protection against symptomatic infection offered by an mRNA booster after a homologous primary series [7,8,10,11]. Findings are more conflicting concerning the potential waning of protection against severe outcomes. After three doses of mRNA vaccine, sustained effectiveness against hospital or intensive care unit (ICU) admission was reported in different studies, including young and elderly individuals within 2 months after booster [8,9,12]. At the same time, another study has demonstrated a significant waning of protection against emergency department visits and hospitalizations 4 months after the third dose [7]. These studies need more data about the medium- and longer-term effectiveness of heterologous schemes and provide limited insights about severe events across age groups. Data on the duration of protection against severe outcomes in boosted individuals by age group are crucial to guide health policies about vaccination programs.

Brazil and Scotland present similarities in vaccination programs (vaccine type used for primary series and booster) and the speed of Omicron spread. Both countries have been offering BNT16b2 or ChAdOx1 as a primary series to all adults and an mRNA booster dose, i.e., BNT16b2 in Brazil and either BNT16b2 or mRNA-1273 in Scotland. Additionally, these

Scottish Government Chief Medical Officer's COVID-19 Advisory Group and its Standing Committee on Pandemics; he is also a member of the UK Government's New and Emerging Respiratory Virus Threats Risk Stratification Subgroup and a member of AstraZeneca's Thrombotic Thrombocytopenic Taskforce. All roles are unremunerated. All other authors declare no competing interests.

**Abbreviations:** CI, confidence interval; COVID-19, Coronavirus Disease 2019; ICU, intensive care unit; OR, odds ratio; RT-PCR, reverse transcriptase polymerase chain reaction; SARS-CoV-2, Severe Acute Respiratory Syndrome Coronavirus 2; TND, test-negative design; VE, vaccine effectiveness; VOC, variant of concern.

countries have reported a rapid surge and dominance of the Omicron variant [13]. The similarities in vaccine administration between Brazil and Scotland, coupled with essential differences in several potential confounders (such as age structure, the timing of delivery to different age groups, and healthcare characteristics such as access to hospitals), offer an opportunity to undertake robust national analyses on the duration of VE during the Omicron era. We aimed to assess the extent and duration of protection against Omicron-associated symptomatic infection and severe outcomes (i.e., COVID-19 hospitalization and death) after an mRNA booster dose in individuals of different age groups who received either BNT16b2 or ChAdOx1 for their primary vaccination series.

## Methods

### Study design, population, and data sources

This study is reported following the Strengthening the Reporting of Observational Studies in Epidemiology (STROBE) guideline (S2 Appendix). We undertook a test-negative design (TND) case–control study to estimate VE for protection against symptomatic infection and severe COVID-19 outcomes. TND is a type of case–control study that uses population test results, with the positive tests being the cases and the negative tests being the controls. It is ideally suited to situations where not everyone in a population is being tested because the factors that influence being tested will apply to both those who tested positive and those who tested negative [14]. Cases were defined as symptomatic individuals with a positive test (reverse transcriptase polymerase chain reaction (RT-PCR) or lateral flow for Brazil and RT-PCR for Scotland) and controls as symptomatic individuals with a negative test. In both countries, symptoms were assessed by self-report. Only the first positive test during the study period was included for each case, and for controls, only the first negative test was included. Controls included individuals with no record of a positive test during the study period.

All individuals aged 18 years or older who reported acute respiratory illness symptoms and tested for Severe Acute Respiratory Syndrome Coronavirus 2 (SARS-COV-2) infection between January 1, 2022, and April 23, 2022, in Brazil and Scotland were eligible for the study (Fig B in S1 Appendix). We excluded the following: (i) individuals who received different vaccines for the second dose from the first; (ii) individuals whose time interval between the first and second doses was less than 14 days; (iii) individuals with less than 115 days between the second and booster doses (therefore a deviation from the official recommendation for Brazil); (iv) tests with missing information of age, sex, city of residence or sample collection date; and (v) sample specimen collection more than 10 days after symptoms onset.

The data in Brazil came from three deterministically linked national structured administrative databases provided by the Ministry of Health: COVID-19 Vaccination Campaign (SI-PNI); Acute Respiratory Infection Suspected Cases (e-SUS-Notifica); and Severe Acute Respiratory Infection/Illness (SIVEP-Gripe). All COVID-19 vaccine doses in Brazil are provided free of charge by the Ministry of Health. All suspected and confirmed cases of COVID-19 must be reported in the e-SUS-Notifica. Regardless of etiology, all severe acute respiratory illness cases must be notified in the SIVEP surveillance system. Therefore, these three databases should provide 100% countrywide coverage of all reported cases [15]. No detailed definition of comorbidities and race in these databases was provided.

The data in Scotland came from the EAVE II platform that brings together datasets on 5.4 million people providing around 99% countrywide coverage [5]. This platform deterministically linked multiple national datasets, including primary healthcare, laboratory, SARS-CoV-2 testing, vaccinations, death, and secondary care data. In both countries, data were anonymized and hosted within secure analytical environments previously described [5].

## Exposures, confounders, and outcomes

The primary exposure was the administration of a COVID-19 vaccine booster dose. The vaccines considered in this study were a homologous series (first and second dose) of either BNT16b2 or ChAdOx1 and an mRNA booster dose (BNT16b2 in Brazil; BNT16b2 or mRNA-1273 in Scotland). We classified exposure in periods as time-varying to allow us to assess waning, stratified by primary series and booster type. In both countries, the exposure periods were unvaccinated, first dose (0 to 13 days, 14 days to 1 month, 2 to 5 months, $\geq$6 months), second dose (0 to 13 days, 14 days to 1 month, 2 to 4 months, $\geq$ 5 months), and a booster dose (0 to 13 days, 14 to 29 days, 1 month, 2 months, 3 months, and $\geq$ 4 months).

We used the unvaccinated individuals as the reference group in both countries to estimate VE. For both countries, we adjusted for the following potential confounders: age (5-year bands), sex, and socioeconomic position (Brazil: Brazilian Municipality Deprivation Index; Scotland: Scottish Index of Multiple Deprivation, both deprivation indexes used national cut-offs to define the quintiles), number of medical comorbidities (Brazil-Diabetes Mellitus, obesity, immunosuppression, chronic respiratory disease, cardiac disease, and chronic kidney disease) or number and types of comorbidities commonly associated with COVID-19 illness based on the QCOVID risk group (Scotland) (Table B in S1 Appendix) [16], state of residence (Brazil) or geographic area (Scotland), and previous infection (none, 3 to 5 months ago, 6 to 12 months, and >1 year), calendar time was included as the week of sample collection, and healthcare worker in Brazil and number of previous RT-PCR in Scotland (as a proxy for healthcare worker). In Scotland, we additionally adjusted for the residential settlement type (6 categories, from large urban to small remote rural areas) and household size. QCOVID risk groups are characteristics used in the QCOVID algorithm to predict the risk of hospital admission and death due to COVID-19 [16].

The two outcomes of interest were symptomatic SARS-CoV-2 infection. In Brazil, severe COVID-19 cases were defined as COVID-19 hospital admission or death. COVID-19 hospitalization was described as a positive specimen being collected up to 14 days before to 3 days after the hospital admission; cases of COVID-19 death were defined as death occurring within 28 days of the positive sample collection date. In Scotland, severe confirmed COVID-19 cases were defined as admissions to hospital or death within 28 days following a positive specimen where COVID-19 was explicitly mentioned in any place on the admission record or death certificate. Severe COVID-19 cases were defined as either (1) confirmed severe COVID-19 cases; (2) any admission to hospital within 14 days after or up to 2 days before a positive test; or (3) any death within 28 days following a positive specimen and so is a broader definition than confirmed severe cases. In our analysis, we used severe confirmed COVID-19 cases in Scotland.

## Statistical analysis

The prospective statistical plan is provided in the S3 Appendix. We applied binomial logistic regression to estimate the odds ratio (OR) and the associated 95% confidence intervals (CIs) of vaccination in cases compared to controls. VE was defined as $(1 - OR) * 100$. Analyses were conducted by primary vaccine series type (BNT16b2 or ChAdOX1) and age group (18 to 49, 50 to 64, $\geq$65) in both countries. In Scotland, we also stratified by mRNA booster type (BNT16b2, or mRNA-1273); Brazil's only mRNA vaccine was BNT162b2. In Scotland, we did not stratify by age group in the analysis of severe outcomes due to the small number of events. In Brazil, only individuals vaccinated with ChAdOx1 as primary series were analyzed in the $\geq$65 years age group due to the relatively small numbers of vaccinated individuals with BNT162b2 in this group.

For sensitivity analysis, we repeated the models used in the principal analysis using the individuals with a second dose without a booster dose as the comparison group (instead of the

unvaccinated group), including a term of the month of the second dose to control for the potential waning of effectiveness. We also performed an exploratory analysis in Brazil using only individuals with a previously confirmed infection to assess the possible under-ascertainment bias of past infections. All $p$-values are two-sided and determined through Wald test. All analyses were undertaken within secure analytical environments, and the analyses were carried out using R statistical software (versions 3.6.1 and 4.1.1).

## Ethical and other approvals

For Brazil, ethics approvals were obtained from the Brazilian National Commission in Research Ethics (CONEP approval number: 4.921.308). The National Research Ethics Service Committee in Scotland, Southeast Scotland 02 (reference number: 12/SS/0201) and Public Benefit and Privacy Panel for Health and Social Care (reference number: 1920–0279) approved the study.

## Results

From January 01, 2022, to April 23, 2022, 5,832,210 individuals (Brazil: 5,276,385; Scotland: 555,825) were analyzed, and the median age and sex ratio were similar among cases and controls (Table 1 and Table A in S1 Appendix). Most of the tests performed in Brazil during the study period were positive (3,009,052; 57.0%), slightly elevated in unvaccinated individuals (203,964; 61.7%) and vaccinated (2,805,088; 56.7%). In Scotland, 352,015 (63.3%) tests were positive, more frequent in unvaccinated individuals (38,988; 74.6%) than in those vaccinated (313,027; 62.2%) (Table 1 and Tables A, C, and D in S1 Appendix).

## Compared to unvaccinated individuals

In Brazil, a primary course of ChAdOx1 and a BNT16b2 booster (ChAdOx1-BNT162b2) dose led to an estimated VE against symptomatic infection of 51.6% (95% CI [51.0, 52.2], $p < 0.001$) after 14 to 29 days, waning to 4.2% (95% CI [0.7, 7.6], $p = 0.02$) at ≥4 months after the booster dose. The VE of a primary course of BNT162b2 and a BNT162b2 booster (BNT162b2-BNT162b2) was 44.6% (95% CI [43.4, 45.8], $p < 0.001$) at 14 to 29 days, waning to −11.8% (95% CI [−35.9, 8.0], $p = 0.26$) at ≥4 months past the booster dose. (Fig 1 and Table E in S1 Appendix).

The estimates of VE against severe outcomes of ChAdOx1-BNT162b2 peaked in Brazil at 93.5% (95% CI [93.0, 94.0], $p < 0.001$) at 14 to 29 days and 82.3% (95% CI [79.7, 84.7], $p < 0.001$) after ≥4 months. The VE of a BNT162b2-BNT162b2 was 92.7% (95% CI [91.0, 94.0], $p < 0.001$) at 14 to 29 days and 74.1% (95% CI [9.1, 92.6], $p = 0.03$) at ≥4 months (Fig 2 and Table F in S1 Appendix).

Scotland had a similar pattern for both booster vaccines (BNT162b2 or mRNA-1273). After the primary series of ChAdOx1, the VE peak at 98.3% (95% CI [87.3, 99.8], $p < 0.001$) 14 to 29 days past booster dose for ChAdOx1-BNT162b2 and 94.4% (95% CI [87.7, 97.5], $p < 0.001$) for ChAdOx1-mRNA-1273 in the same period, declining to 77.8% (95% CI [51.4, 89.9], $p < 0.001$) and 76.0% (95% CI [59.3, 85.9], $p < 0.001$) at ≥4 months, respectively. (Fig 2 and Table F in S1 Appendix).

## Effectiveness by age group

In Brazil, the VE against symptomatic infection of ChAdOx1-BNT162b2 and BNT162b2-BNT162b2 presented values close to 50% in all age groups at 14 to 29 days but declined more sharply in the younger (18 to 49 years), with no protection in this age group ≥4

**Table 1. Characteristics of individuals tested for SARS-CoV-2 in Brazil and Scotland.**

| Characteristic–n(%) | Brazil | | Scotland | |
| --- | --- | --- | --- | --- |
| | **Cases, N = 3,011,812** | **Controls, N = 2,269,774** | **Cases, N = 352,015** | **Controls, N = 203,810** |
| **Age group–years** | | | | |
| 18–49 | 2,189,355 (72.7) | 1,680,102 (74.0) | 244,517 (69.5) | 136,390 (66.9) |
| 50–64 | 692,693 (23.0) | 494,404 (21.8) | 82,904 (23.6) | 50,109 (24.6) |
| ≥ 65 | 129,764 (4.3) | 95,268 (4.2) | 24,594 (7.0) | 17,311 (8.5) |
| **Sex-Female** | 1,754,321 (58.2) | 1,342,774 (59.2) | 196,485 (55.8) | 125,056 (61.4) |
| **Test type** | | | | |
| Antigen | 2,458,769 (81.6) | 1,954,203 (86.1) | - | - |
| RT-PCR | 553,043 (18.4) | 315,571 (13.9) | 352,015 (100.0) | 203,810 (100.0) |
| **No. comorbidities or QCOVID Risk** | | | | |
| 0 | 2,755,169 (91.5) | 2,046,435 (90.2) | 226,394 (64.3) | 122,253 (60.0) |
| 1 | 206,214 (6.8) | 178,454 (7.9) | 94,752 (26.9) | 58,481 (28.7) |
| 2 | 42,373 (1.4) | 37,576 (1.7) | 23,396 (6.7) | 16,809 (8.2) |
| ≥3 | 8,056 (0.3) | 7,309 (0.3) | 6,832 (2.0) | 6,267 (3.1) |
| **Deprivation Index** | | | | |
| 1 (Least) | 939,579 (31.2) | 792,447 (34.9) | 65,536 (18.6) | 42,071 (20.6) |
| 2 | 560,955 (18.6) | 407,162 (17.9) | 67,242 (19.1) | 41,989 (20.6) |
| 3 | 589,431 (19.6) | 425,703 (18.8) | 65,183 (18.5) | 38,394 (18.8) |
| 4 | 581,790 (19.3) | 388,683 (17.1) | 74,900 (21.3) | 40,838 (20.0) |
| 5 (Most) | 339,183 (11.3) | 255,181 (11.2) | 79,154 (22.5) | 40,518 (19.9) |
| (Missing) | 874 (0.0) | 598 (0.0) | - | - |
| **Previous SARS-CoV-2 Infection** | | | | |
| Not | 2,808,497 (93.2) | 2,020,509 (89.0) | 316,843 (90.0) | 167,041 (82.0) |
| 3–6 months ago | 9,871 (0.3) | 16,766 (0.7) | 10,874 (3.1) | 15,345 (7.5) |
| 6–12 months ago | 106,687 (3.5) | 134,347 (5.9) | 12,274 (3.5) | 11,548 (5.7) |
| >1 year ago | 86,757 (2.9) | 98,152 (4.3) | 12,024 (3.4) | 9,876 (4.8) |
| **Vaccination Status** | | | | |
| One dose–ChAdOx1 | 90,069 (3.0) | 61,823 (2.7) | 2,307 (0.6) | 1,026 (0.5) |
| One dose–BNT162b2 | 120,949 (4.0) | 111,999 (4.9) | 8,193 (2.3) | 3,021 (1.5) |
| Two doses–ChAdOx1 | 1,203,551 (40.0) | 696,531 (30.7) | 27,908 (7.9) | 10,807 (5.3) |
| Two doses–BNT162b2 | 851,007 (28.3) | 711,679 (31.4) | 62,285 (17.7) | 26,746 (13.1) |
| Three doses–BNT162b2 | 69,628 (2.3) | 87,694 (3.9) | 106,588 (30.3) | 75,415 (37.0) |
| Three doses–ChAdOx1 | 472,644 (15.7) | 473,369 (20.9) | 103,407 (29.4) | 71,399 (35.0) |
| Four doses–ChAdOx1 | - | - | 1159 (0.3) | 683 (0.4) |
| Four doses–BNT162b2 | - | - | 574 (0.1) | 330 (0.1) |
| Unvaccinated | 203,964 (6.8) | 126,679 (5.6) | 38,988 (11.1) | 13,250 (6.5) |
| **Hospitalization** | 38,284 (1.3) | 22,596 (1.0) | - | - |
| **Death** | 12,270 (0.4) | 4,985 (0.2) | - | - |
| **Severe outcome** | 40,522 (1.3) | 23,773 (1.0) | 1,375 (0.4) | 372 (0.2) |

RT-PCR, reverse transcriptase polymerase chain reaction; SARS-CoV-2, Severe Acute Respiratory Syndrome Coronavirus 2.

months. However, the relative VE of both ChAdOx1-BNT162b2 and BNT162b2-BNT162b2 exhibited a similar decline in all age groups ≥4 months after the booster dose. In the exploratory analysis using only individuals with a previously confirmed infection, the ChAdOx1-BNT162b2 group presented a VE close to 70% at 14 to 29 days past booster dose in all age groups, declining, to a lesser degree than the primary analysis, to levels close to 20%.

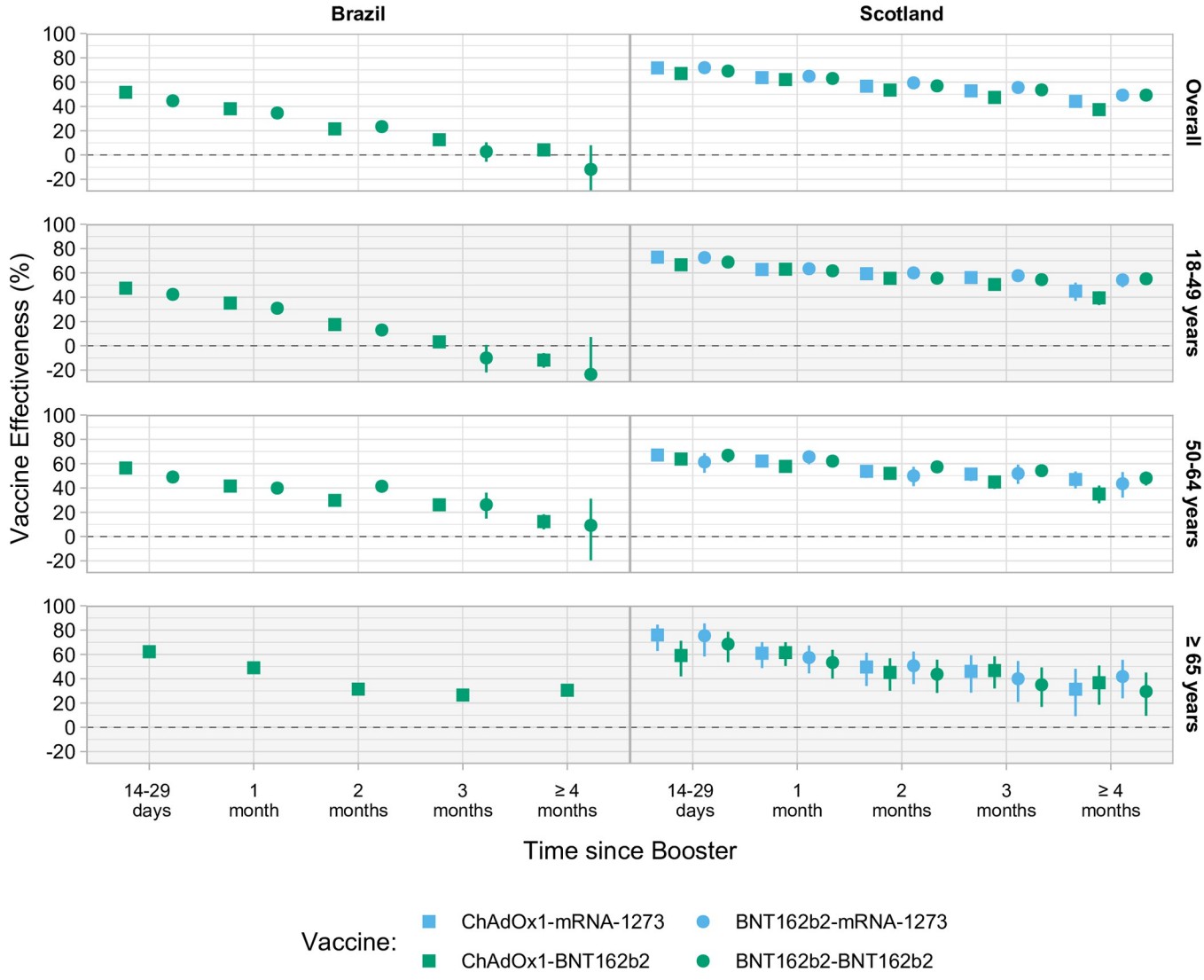

**Fig 1. Estimated VE against symptomatic SARS-CoV-2 infection in Brazil and Scotland, overall and stratified by age group.** The square represents the estimated VE of the booster dose after the primary series with ChAdOx1, and the circle represents the booster dose after the primary series with BNT162b2. Green represents the booster dose with BNT162b2 and blue with mRNA-1273. Error bars represent the 95% Wald CI. CI, confidence interval; SARS-CoV-2, Severe Acute Respiratory Syndrome Coronavirus 2; VE, vaccine effectiveness.

Unlike Brazil, the VE against symptomatic infection in Scotland decreased more in the older groups than the younger ones. (Fig 1 and Table E in S1 Appendix).

The VE of ChAdOx1-BNT162b2 against severe outcomes in all age groups peaked around 90% at 14 to 29 days past and stayed higher than 80% past 4 months after the booster dose. The VE by age group of BNT162b2-BNT162b2 was like the ChAdOx1. (Fig 2 and Table E in S1 Appendix). In Scotland, due to the small number of severe COVID-19 cases, we did not perform the stratified analysis by age group for this outcome.

## Effectiveness of the second booster dose

The analysis of VE of the second booster dose was conducted only in Scotland due to insufficient numbers in Brazil. The VE at 7 to 29 days, after a second booster with BNT162b2, in

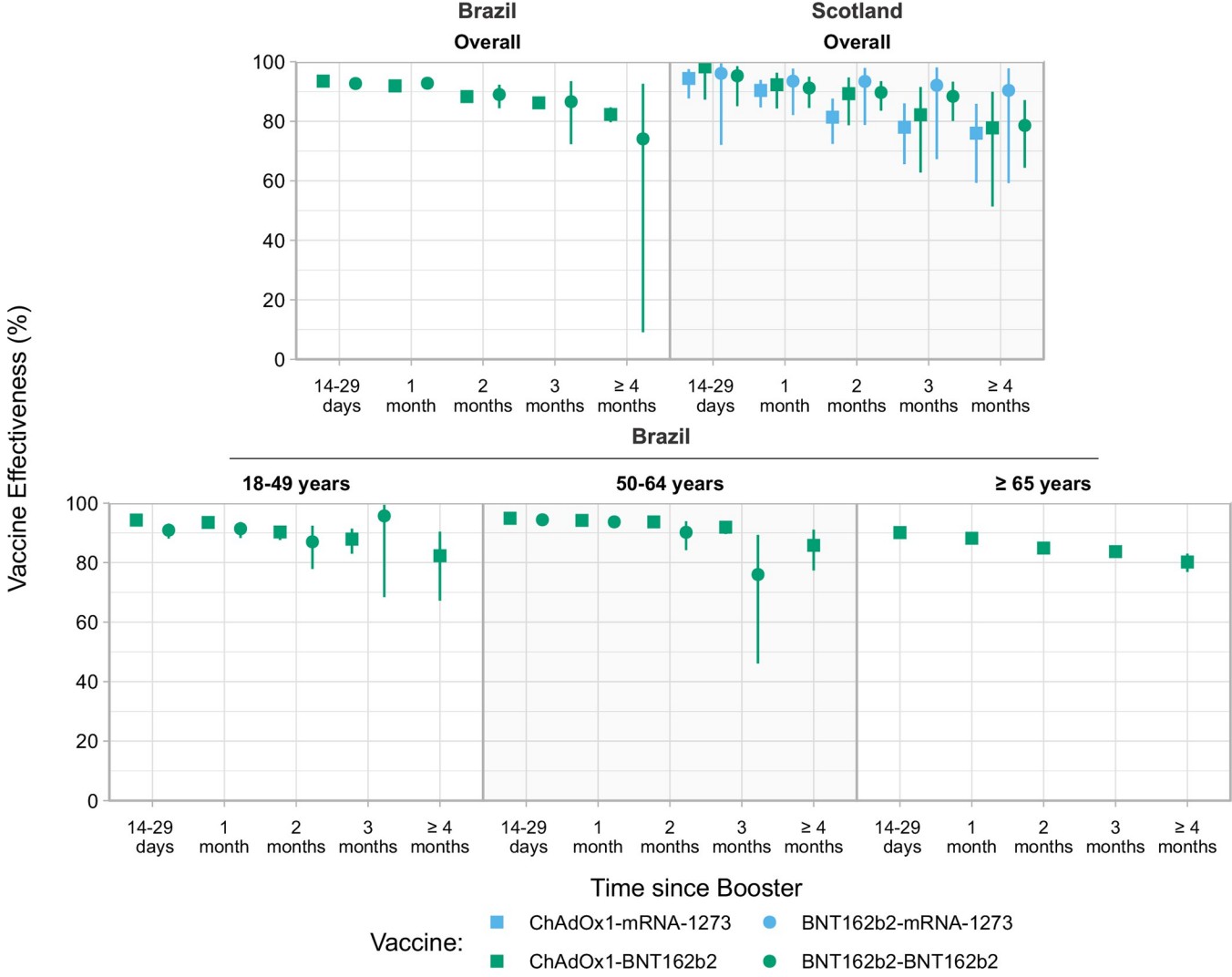

**Fig 2. Estimated VE against severe COVID-19 (hospitalization or death) in Brazil and Scotland, stratified by age groups.** The square represents the estimated VE of the booster dose after the primary series with ChAdOx1, and the circle represents the booster dose after the primary series with BNT162b2. Green represents the booster dose with BNT162b2 and blue with mRNA-1273. The error bar represents the 95% Wald CI. CI, confidence interval; COVID-19, Coronavirus Disease 2019; VE, vaccine effectiveness.

individuals with a primary course of ChAdOx1 was 40.6% (95% CI [20.9, 55.4], $p < 0.001$); in the case of a second booster with mRNA-1273, the VE was 47.5% (95% CI [21.5, 64.9], $p = 0.002$). Regarding individuals with a primary course of BNT162b2, the VE after 7 to 29 days after a second booster was 61.1% (95% CI [43.6, 73.2], $p < 0.001$) in the case of BNT162b2 booster and 69.2% (95% CI [46.5, 82.3], $p < 0.001$) for mRNA-1273 booster. The comparison against individuals with only two doses provided similar results. (Table E in S1 Appendix).

## Compared to individuals with only the primary course (two doses)

In Brazil, the relative VE of ChAdOx1-BNT162b2 compared to individuals with only two doses of ChAdOx1 against symptomatic infection was 50.9% (95% CI [50.4, 51.4], $p < 0.001$) after 14 to 29 days, waning to 7.0% (95% CI [3.8, 10.1], $p < 0.001$) at ≥4 months after the

booster dose. The relative VE of a BNT162b2-BNT162b2 compared to only the primary course with BNT162b2 was 40.5% (95% CI [39.3, 41.7], $p < 0.001$) at 14 to 29 days, waning to 9.7 (95% CI [−9.6, 25.6], $p < 0.001$) at ≥4 months after the booster dose. (Table G in S1 Appendix).

In Brazil, the relative VE against severe outcomes of ChAdOx1-BNT162b2 was 79.0% (95% CI [77.3, 80.6], $p < 0.001$) at 14 to 29 days and waned to 59.4% (95% CI [53.3, 64.7], $p < 0.001$) after ≥4 months. The relative VE of BNT162b2-BNT162b2 was 54.6% (95% CI [43.9, 63.2], $p < 0.001$), and 38.2% (95% CI [−125.0, 83.0], $p = 0.47$) at ≥4 months after the booster dose. The relative VE of ChAdOx1-BNT162b2 peaked around 80% in all age groups, declining to 68.8% (95% CI [43.5, 82.8], $p < 0.001$), 55.8% (95% CI [31.2, 71.7], $p < 0.001$) and 56.2% (95% CI [48.7, 62.5], $p < 0.001$) in the 18 to 49, 50 to 64, and ≥65 years, respectively. (Table H in S1 Appendix).

In Scotland, the relative VE of ChAdOx1-BNT162b2 against symptomatic infection was 63.5% (95% CI [61.4, 64.7], $p < 0.001$) after 14 to 29 days, waning to 28.8% (95% CI [23.8, 33.5], $p < 0.001$) at ≥4 months after the booster dose. The relative VE of a BNT162b2-BNT162b2 was 62.7% (95% CI [61.0, 64.4], $p < 0.001$) at 14 to 29 days, waning to 29.1% (95% CI [24.0, 33.2], $p < 0.001$) at ≥4 months after the booster dose. The schemas with mRNA-1273 exhibited similar values. (Table G in S1 Appendix) The relative VE of booster doses in individuals with ChAdOx1 as primary series against severe outcomes peaked at 85.2% (95% CI [67.3, 93.3], $p < 0.001$) for BNT162b2 booster and 95.5% (95% CI [67.3, 99.4], $p = 0.002$) for mRNA-1273 booster, decreasing to 54.9% (95% CI [14.2, 76.3], $p = 0.02$) for BNT162b2 and 59.3% (95% CI [3.6, 82.8], $p = 0.04$) for mRNA-1273 past 4 months after the booster. A similar pattern was found for individuals with BNT162b2 as primary series but with less precise estimates. (Table H in S1 Appendix).

## Discussion

Following a homologous primary series of BNT16b2 or ChAdOx1 vaccine, mRNA boosters (BNT16b2 or mRNA-1273) provided substantial protection against severe COVID-19 cases during the predominance of Omicron variant for at least 3 months. However, there was only moderate protection against symptomatic infection at 14 to 29 days after the booster dose administration, which sharply decreased by ≥4 months.

Our results are comparable to previous observational studies during the Omicron period, which reported the waning effectiveness of mRNA boosters against symptomatic infection. Protection against infection is modest following the booster dose and quickly decreases 4 weeks post-booster [10,11]. Older individuals experienced a faster waning than younger people [4,17]. However, we found a more pronounced waning against symptomatic infection in the younger age groups in Brazil and a comparable waning in all age groups in Scotland.

The finding of temporary protection against symptomatic infection is consistent with neutralization data, suggesting the need for a third dose to elicit antibodies with neutralizing activity against Omicron and decaying titers over time [18,19]. Neutralizing antibody titers seem to increase again early after a fourth dose, but the duration remains unknown [20]. Observational studies evaluating the effect of vaccination on previously infected individuals reported that antibody levels peaked after three immune stimuli, either by vaccine or infection, without any significant increment after a fourth stimulus [21,22]. On the other hand, cellular immunity seems to stay robust against the Omicron variant after 3 months [23]. It plays a significant role in protecting against severe disease [24,25]. It likely will stay highly effective against variants of the SARS-CoV-2 virus due to the capacity of T cells to still recognize mutated epitopes from SARS-CoV-2 [23,25,26]. Consistent with data on cellular immunity, we observed a slight

waning of protection against severe outcomes in the Omicron period. A fourth dose seems to increase the protection against severe illness but not against infection in individuals aged ≥60 years [27]. We found similar levels of protection against symptomatic infection in individuals with one and two booster doses. Together, these data suggest that while humoral response and VE against infection seem to be highly affected by the Omicron variant, a limited impact is observed on cellular immunity and protection against severe disease.

We analyzed the waning by age group due to the influence of immunosenescence in VE. We observe sustained protection against severe outcomes and a slight decline in protection in individuals aged ≥65 years, represented by the decrease in the last period in both comparisons: against unvaccinated individuals and individuals with only two doses of vaccine. Different age structures may have contributed to the apparently contradictory results observed in some studies that have addressed the duration of protection of mRNA booster dose against severe disease. For example, there was high and sustained protection (over 80% VE) at ≥7 weeks after booster and no evidence of waning in Qatar, a country with only 9% of the population ≥50 years old [8]. In contrast, a study analyzing data from 10 states of the United States of America found evidence of waning protection against hospitalization, with VE dropping from 91% at 2 months to 78% at 4 months of booster dose [7]. In addition to age, follow-up time seems essential in the analysis of waning. In a study performed in Finland only on individuals >70 years old, sustained protection was observed against admission to hospital and ICUs. Still, follow-up after booster was up to 2 months [9], which may at least partially explain the differences between their observed declines and what we found in VE in the elderly. Indeed, we observed the lowest VE against severity for the elderly past 4 months. Those previous studies provided no data about VE across age groups. Thus, differences in age groups and the follow-up time seem to be involved in the differences observed in VE after mRNA booster dose across the studies.

Assessment of vaccine waning from observational studies during the COVID-19 pandemic is methodologically challenging due to dynamic changes during the vaccination program. Challenges include prioritizing vaccine delivery to higher-risk groups, e.g., the elderly, individuals with comorbidities, and healthcare professionals, which led to more time after the booster dose than in other individuals. In addition, different intervals between the last dose in the primary vaccination series and the booster dose have been used in different places. All these factors are compounded by the different sublineages of the Omicron variant circulating during the follow-up period, which can introduce potential bias. However, conducting harmonized analyses in two countries at the national level mitigates the possibility of spurious results driven by unmeasured confounders.

Brazil and Scotland present several differences, such as variability in vaccination programs, circulating subvariants of Omicron, and population characteristics such as age structure, testing policy, and vaccination status. For example, up to April, Brazil had vaccinated less than 50% of the eligible population for booster doses, while in Scotland, this index was more than 85%. These aspects may have influenced the lower VE against infection in Brazil compared to Scotland. However, despite all dissimilarities between countries, a similar pattern of quickly VE waning against infection and durable protection against severe disease was observed, reinforcing the robustness of these findings.

To our knowledge, the present study is the largest to investigate the waning of mRNA booster doses against severe outcomes in the Omicron era. In both countries, similar results were obtained using different reference groups to assess vaccine protection: unvaccinated individuals in the main analysis and individuals with only two vaccine doses in the sensitivity analyses. We adjusted for several clinically relevant factors by deterministically linking various national clinical databases. Using a TND case–control study, we mitigated the risk of bias due

to differences in health-seeking behavior between vaccinated and unvaccinated groups [14,28].

However, there are some limitations to note. First, we assumed all cases in the study period were associated with the Omicron variant. A few cases may have been due to other variants, including Delta. To mitigate this limitation, we restricted our analyses to the period when Omicron was dominant in both countries. Second, most of the tests performed in Brazil during the study period were lateral flow, which may induce bias in the results due to misclassifying cases as controls. Third, we were unable to discriminate between Omicron lineages in our study. Nevertheless, initial studies have suggested VE against Omicron subvariants to be similar [8,29,30]. Fourth, as in any observational study, residual confounding might exist. However, in previous studies, the adjustment for the chosen cofounders provided demonstrable control for bias. We cannot exclude that bias could arise from the unexpected effects of COVID-19 vaccines in other acute respiratory illnesses, protecting these individuals. Fifth, we have defined COVID-19-associated hospitalization as any admission episode that occurs within 14 days of a positive SARS-CoV-2 test or positive test within 72 hours of hospital admission. Consequently, a subset of hospitalizations may be composed of incidental cases. However, although possible in Scotland, such incidental cases are unlikely to occur in Brazil. The Brazilian hospitalization database used in the present study only includes individuals with severe acute respiratory syndrome symptoms tested for SARS-CoV-2 infection. Sixth, there are a disproportional number of individuals in our sample from Brazil's first quintile of deprivation (low deprivation). It is likely due to the use of the municipality deprivation index, indicating that cities with lower deprivation maintain a broader testing policy than cities with higher deprivation. Seventh, asymptomatic and mild infections may have been underrecognized if previously infected individuals were not tested. In Brazil, we found negative VE against symptomatic infection in the younger group in the last period past booster. However, in the exploratory analysis using only individuals with previously confirmed SARS-CoV-2 infection, in this scenario, the result was not replicated, indicating a possible bias of under-ascertainment of previous infection in the younger individuals, with more unvaccinated individuals with previous undetected infection.

In summary, our study has shown that older individuals are at the highest risk of experiencing severe outcomes after infection with the Omicron variant, even after receiving a booster dose. The recommendation for a second mRNA booster dose, now being implemented for these groups in several countries, seems sensible for preventing severe forms of COVID-19. Our data indicate that infection prevention and, thus, community protection may not be a realistic target with currently available vaccines. The durability of protection against hospitalization and death remains an open question underscoring the need for studies with extended follow-up periods. New vaccines, mainly those aimed at interrupting transmission, possibly by enhancing mucosal immunity, are necessary to reduce the risk of Omicron infection and transmission.

## Supporting information

**S1 Appendix. Additional tables and figures. Table A.** Additional population characteristics. **Table B.** Conditions of QCovid risk algorithm. **Table C.** Vaccination status of individuals tested for SARS-CoV-2 in Brazil (A) and Scotland (B), according to the test result and severity of disease. Vaccinees data were detailed according to the time after each dose. **Table D.** Time interval in days—median (interquartile interval)—between vaccination and test in Brazil and Scotland, according to the type of vaccine used in the primary series. **Table E.** Vaccine effectiveness against symptomatic infection in Brazil and Scotland, expressed in percentages (95% CI), according to the type of vaccine used at the primary series and by age group. Reference

group: individuals unvaccinated. **Table F.** Vaccine effectiveness against severe outcomes in Brazil and Scotland, expressed in percentages (95% CI), according to the type of vaccine used at the primary series and by age group. Reference group: individuals unvaccinated. **Table G.** Relative vaccine effectiveness against symptomatic infection in Brazil and Scotland expressed in percentages (95% CI) according to the type of vaccine used at the primary series and by age group. Reference group: individuals that received only a primary series. **Table H.** Relative vaccine effectiveness against severe disease in Brazil and Scotland expressed in percentages (95% CI) according to the type of vaccine used at the primary series and by age group. Reference group: individuals that received only a primary series. **Table I.** Vaccine effectiveness against symptomatic infection among individuals with a previously confirmed infection in Brazil compared to unvaccinated. Results were reported as percentages (95% CI), according to the type of vaccine used in the primary series and by age group. **Fig A.** Distribution of variants of concern in Brazil and Scotland over time. Brazil (A) and Scotland (B). **Fig B.** STROBE flowchart of the study population in Brazil (A) and Scotland (B). **Fig C.** Distribution case and control over time in each country for individuals unvaccinated or vaccinated with ChAdOx1 or BNT162b2 as primary series. Brazil (A) and Scotland (B). **Fig D.** Uptake of each dose, including booster dose, in individuals vaccinated with ChAdOx1 or BNT162b2 as primary series. Vaccination numbers in Brazil (A) and Scotland (B), stratified by age group and primary series. Different y-axis scales in each age group.
(DOCX)

**S2 Appendix. STROBE/RECORD checklist.**
(DOCX)

**S3 Appendix. Statistical analysis plan.**
(DOCX)

## Acknowledgments

This study is part of the Fiocruz VigiVac program, and the authors acknowledge DATASUS for its diligent work in providing the unidentified Brazilian databases. GLW, MLB, VSB, and MB-N are Brazilian National Research Council research fellows. SVK acknowledges funding from an NRS Senior Clinical Fellowship (SCAF/15/02), the Medical Research Council (MC_UU_00022/2), and the Scottish Government Chief Scientist Office (SPHSU17). ESP acknowledges funding from the Wellcome Trust (213589/Z/18/Z). T.C.-S. is a PhD student at the Post-Graduation Program in Health Sciences-UFBA, which is supported by the Coordenação de Aperfeiçoamento de Pessoal de Nível Superior-Brasil, finance code 001. We thank Dave Kelly from Albasoft (Inverness, UK) for his support with making primary care data available; Wendy Inglis-Humphrey, Vicky Hammersley, and Laura Brook (University of Edinburgh, Edinburgh, UK); and Pam McVeigh, Amanda Burridge, and Afshin Dastafshan (Public Health Scotland, Glasgow, UK) for their support with project management and administration.

## Author Contributions

**Conceptualization:** Chris Robertson, Srinivasa Vittal Katikireddi, Aziz Sheikh, Manoel Barral-Netto.

**Data curation:** Vinicius de Araujo Oliveira.

**Formal analysis:** Thiago Cerqueira-Silva, Chris Robertson.

**Funding acquisition:** Aziz Sheikh, Manoel Barral-Netto.

**Investigation:** Thiago Cerqueira-Silva, Chris Robertson, Srinivasa Vittal Katikireddi, Viviane S. Boaventura.

**Methodology:** Thiago Cerqueira-Silva, Chris Robertson, Srinivasa Vittal Katikireddi, Neil Pearce.

**Project administration:** Aziz Sheikh, Manoel Barral-Netto.

**Resources:** Vinicius de Araujo Oliveira, Manoel Barral-Netto.

**Software:** Juracy Bertoldo Junior.

**Supervision:** Viviane S. Boaventura, Aziz Sheikh, Manoel Barral-Netto.

**Writing – original draft:** Thiago Cerqueira-Silva, Syed Ahmar Shah, Mauro Sanchez, Viviane S. Boaventura.

**Writing – review & editing:** Thiago Cerqueira-Silva, Syed Ahmar Shah, Srinivasa Vittal Katikireddi, Enny S. Paixão, Igor Rudan, Gerson O. Penna, Neil Pearce, Guilherme Loureiro Werneck, Mauricio L. Barreto, Viviane S. Boaventura, Aziz Sheikh, Manoel Barral-Netto.

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
