## [Editor Report · Decision Letter 0]

19 May 2022

Dear Dr Cerqueira-Silva, 

Thank you for submitting your manuscript entitled "Waning of mRNA boosters after homologous primary series with BNT162b2 or ChAdOx1 against symptomatic infection and severe COVID-19 in Brazil and Scotland: a test-negative design case-control study" for consideration by PLOS Medicine.

Your manuscript has now been evaluated by the PLOS Medicine editorial staff and I am writing to let you know that we would like to send your submission out for external peer review.

Please re-submit your manuscript within two working days, i.e. by May 23 2022 11:59PM.

Kind regards,

Caitlin Moyer, Ph.D.

Associate Editor

PLOS Medicine

---

## [Decision Letter · Decision Letter 1]

13 Jul 2022

Dear Dr. Cerqueira-Silva,

Thank you very much for submitting your manuscript "Waning of mRNA boosters after homologous primary series with BNT162b2 or ChAdOx1 against symptomatic infection and severe COVID-19 in Brazil and Scotland: a test-negative design case-control study" (PMEDICINE-D-22-01640R1) for consideration at PLOS Medicine. 

Your paper was evaluated by a senior editor and discussed among all the editors here. It was also discussed with an academic editor with relevant expertise, and sent to three independent reviewers, including a statistical reviewer. The reviews are appended at the bottom of this email and any accompanying reviewer attachments can be seen via the link below:

[LINK]

In light of these reviews, I am afraid that we will not be able to accept the manuscript for publication in the journal in its current form, but we would like to consider a revised version that addresses the reviewers' and editors' comments. Obviously we cannot make any decision about publication until we have seen the revised manuscript and your response, and we plan to seek re-review by one or more of the reviewers. 

We expect to receive your revised manuscript by Aug 03 2022 11:59PM. Please email us (plosmedicine@plos.org) if you have any questions or concerns.

We look forward to receiving your revised manuscript. 

Sincerely,

Caitlin Moyer, Ph.D.

Associate Editor

PLOS Medicine

plosmedicine.org

From the Academic Editor:

The vaccine effectiveness literature is rapidly evolving. The benefits of a single booster are well characterised in the literature and the policy discussion now seems to be more focussed on whether a second booster should be provided to more populations (per WHO HCWs, elderly, and immunocompromised are already eligible for an additional booster, https://www.who.int/news/item/17-05-2022-interim-statement-on-the-use-of-additional-booster-doses-of-emergency-use-listed-mrna-vaccines-against-covid-19). Therefore, one suggestion was to see whether the authors can use their Brazilian and Scottish data to look at the impact of the fourth dose. Moreover, the primary vaccine series of ChAdOx1+mRNA booster is unique to specific countries and so I think teasing out the impact of this vaccine combination is worthwhile (currently they have combined it with primary mRNA vaccination series which can confound effects). Please find these suggestions below:

Major:

1. Existing literature (see Effectiveness of COVID-19 vaccines against SARS-CoV-2 variants of concern: a systematic review and meta-analysis. Zeng B, Gao L, Zhou Q, Yu K, Sun F. BMC Med. 2022 May 23;20(1):200) has shown that ChAdOx1 does not perform the same as BNT162b2. Due to potentially introducing confounding by primary vaccination series, please do not group these two together and compare primary vaccination series of BNT162b2+mRNA booster and primary vaccination series of ChAdOx1+mRNA booster with unvaccinated. On the same note in lines 292-307, please contextualise discussion of other studies by the primary vaccination series + booster in each country/study rather than exclusively grouping by number of vaccination doses.

2. The 13 week threshold used is unorthodox and inconsistent with other studies. As suggested by one of the reviewers, agree extending out to 6 months and evaluating greater than or equal to 6 months as the last waning threshold makes sense. If this is not possible at least use greater than or equal to 3 months (or 12 weeks).

3. Given the benefits of a fourth dose for specific populations (HCWs, elderly, immunocompromised (https://www.who.int/news/item/17-05-2022-interim-statement-on-the-use-of-additional-booster-doses-of-emergency-use-listed-mrna-vaccines-against-covid-19). Would suggest the authors consider updating the analysis to also include a second booster dose for whatever periods of follow-up and populations are available.

Minor:

4. The Qatari study has now been published and can be updated. Chemaitelly H, Ayoub HH, Al Mukdad S,et al. Duration of mRNA vaccine protection against SARS-CoV-2 Omicron BA.1 and BA.2 subvariants in Qatar. Nat Commun. 2022 Jun 2;13(1):3082.

5. In the limitations discussion sublineages can also add BA4 and BA 5 now.

Other Editorial Points:

6. Title: We suggest revising the title for clarification. Please revise your title according to PLOS Medicine's style. Your title must be nondeclarative and not a question. It should begin with main concept if possible. "Effect of" should be used only if causality can be inferred, i.e., for an RCT. Please place the study design ("A randomized controlled trial," "A retrospective study," "A modelling study," etc.) in the subtitle (ie, after a colon).

7. Data availability statement: Please provide the weblink/email contact information for requests to access the data from Scotland.

8. Abstract: Please structure your abstract using the three PLOS Medicine headings (Background, Methods and Findings, Conclusions). Please combine the Methods and Findings sections into one section, “Methods and findings”.

9. Abstract: Methods and Findings: Please describe the national databases, in terms of the populations represented and numbers of participants included, as well as the time period of the study, the maximum length of follow up, and how the main outcome measures were defined/quantified. We suggest including information similar to what is reported at lines 115-117 of the Methods (“All individuals aged 18 years or older who reported acute respiratory illness symptoms and tested for SARS-CoV-2 infection between January 1, 2022 and March 7, 2022 for Brazil and December 20, 2021 and March 9, 2022 for Scotland were eligible for the study”).

10. Abstract: Methods and Findings: Please report p values as p<0.001 where applicable.

11. Abstract: Methods and Findings: Please include a summary of adverse events if these were assessed in the study.

12. Abstract: Methods and Findings: In the last sentence of the Abstract Methods and Findings section, please describe the main limitation(s) of the study's methodology.

13. Abstract: Conclusions: Please address the study implications without overreaching what can be concluded from the data; the phrase "In this study, we observed ..." may be useful.

14. Author summary: At this stage, we ask that you include a short, non-technical Author Summary of your research to make findings accessible to a wide audience that includes both scientists and non-scientists. The Author Summary should immediately follow the Abstract in your revised manuscript. This text is subject to editorial change and should be distinct from the scientific abstract. Please see our author guidelines for more information: https://journals.plos.org/plosmedicine/s/revising-your-manuscript#loc-author-summary

15. Main text: Please place in-text citations of references within brackets, before the sentence punctuation.

16. Methods: For all observational studies, in the manuscript text, please indicate: (1) the specific hypotheses you intended to test, (2) the analytical methods by which you planned to test them, (3) the analyses you actually performed, and (4) when reported analyses differ from those that were planned, transparent explanations for differences that affect the reliability of the study's results. If a reported analysis was performed based on an interesting but unanticipated pattern in the data, please be clear that the analysis was data-driven.

17. Methods: Line 145: Please explain “comorbidities” and please describe all factors that were adjusted for, and explain how these data were obtained.

18. Methods: Line 151: Please describe QCovid risk group here.

19. Methods: Line 165-166: Please clarify if this refers to weeks: “A trend test, evaluating the change in estimate between 2-4, 5-8, 9-12, and ≥13 past booster dose…”

20. Methods: Statistical analyses: For analyses where a p value is given, please specify the statistical test used to determine it as well as the significance level used (eg, P<0.05, two-sided).

21. Methods: Please ensure that the study is reported according to the STROBE guideline. Thank you for including the completed STROBE checklist as Supporting Information.

Please add the following statement, or similar, to the Methods: "This study is reported as per the Strengthening the Reporting of Observational Studies in Epidemiology (STROBE) guideline (S1 Checklist)."

When completing the checklist, please use section and paragraph numbers throughout to indicate locations within the text.

22. Methods: Did your study have a prospective protocol or analysis plan? Please state this (either way) early in the Methods section.

23. Results: Line 238-248: Please also provide p values for these results, where applicable.

24. Discussion: Please present and organize the Discussion as follows: a short, clear summary of the article's findings; what the study adds to existing research and where and why the results may differ from previous research; strengths and limitations of the study; implications and next steps for research, clinical practice, and/or public policy; one-paragraph conclusion.

25. Discussion: Line 267: Please clarify “severe Omicron COVID-19 outcomes” to indicate that this means Omicron era cases.

26. Figures and Tables: Please provide titles and legends for all figures (including those in Supporting Information files). Please check that any abbreviations used within Tables and Figures is completely defined in the legend. Please consider avoiding the use of red and green together in the same figure in order to make your figure more accessible to those with color blindness.

27. Figure 1 and Figure 2: Please indicate in the figure caption the meaning of the error bars. Please indicate in the legend if severe COVID-19 indicates both hospitalization and death.

28. S1 Table: Please describe in the Methods how characteristics of cases and controls were evaluated/reported. Please describe how characteristics (e.g. obesity/immunosuppression/respiratory disease/kidney disease/cardiac disease etc) were determined. How was race/ethnicity defined and by whom? In S1 part B, please use variable names under “characteristic” that are more intuitive and/or similar to S1 part A.

29. S2 Table: Please note all values are n(%).

30. S3 Table: Please include at the top that the categories represent ages. In Part B (Scotland, severe outcomes): In the legend please define cells with “*” indicated.

31. S4 Table, S5 Table: Where p values are reported, please report as p<0.001 where applicable. When p<0.01 please report to 3 decimal places, and when p>/=.01 please report to two decimal places. Please indicate that the table is presented by age category. In the legend, please define cells with “-” reported.

32. S6 Table: Please label the columns at the top with “age” and “primary series” as appropriate. In the legend for Part B, please define the cells with “*” reported.

33. S1 Appendix: There seems to be an unlabelled table between S6 Table and S7 Table, titled “Severe Outcomes” and reporting VE for each primary series type.

34. S7 Table: Part A: The right-hand side of the table is cut off.

Comments from the reviewers:

Reviewer #1: See attachment

Michael Dewey

Reviewer #2: 

Review Cerqueira-Silva et al, PlosMed

The strength of this study is that it takes place in two countries, with different populations but similar vaccination strategies, among large populations. The design is solid and well-execulted. I have two high-level critiques. First, is that they call this a study of waning, even in the title, but only follow-up persons to 13+ weeks--- not sure exactly how long in that furthest follow-up group. This relatively short follow-up time is sufficient for symptomatic infection with omicron where waning happens swiftly, but probably not enough followup time for severe disease. 2. The authors overstate the amount of waning after booster among persons >=65 years. Their results show a minimal amount of decrease in VE (<10 percentage points) with overlapping CI< yet they lead off their discussion by highlighting the waning in this group. 

Specific comments.

Line 94. Delete « offered »

Line 112. Lateral flow tests have lower sensitivity. Most tests in Brazil were lateral flow (Table 1) Authors could mention this as a potential limitation, due to misclassification of case status. Do they have test characteristics for the lateral flow tests used? 

Line 142. What is vaccination coverage in general populations? If the unvaccinated percentage is small (<10%) it could suggest a potential bias among the remaining unvaccinated population in terms of their risk of exposure. From table 1, it seems that this is <10%, so this should be mentioned as a potential bias in the discussion (if not already).

Line 158. Given the matched enrollment in Scotland, how were the matched groups identified in the analysis. Why not conditional logistic regression?

Line 166. Missing "weeks"?

Line 170. Note that this sensitivity analysis will calculate a relative VE rather than an absolute VE. If there is still residual VE from two doses, then the relative Ve will be lower than absolute VE. More an issue for severe disease. it should be stated that this analysis yields a relative VE. This is seen in table S7 especially in Scotland.

Line 186. Replace word "and" with "compared with"??

Line 188. The median followup time since vaccination / booster should be given. In the >13 week group, what is the median folllowup time in that group. If it is not much more than 13 weeks, it is not very long.

Table 1. somewhat surprising that there are a greater percentage in the highest deprivation index categories in Scotland than Brazil. Why? Use of different deprivation indices? 

Line 220, figures. The VE in Brazil was consistently higher, including by age group. Why do the authors think this is the case? They should comment on that. Although table 1 shows low prior infection in both countries, one wonders whether seroprevalence in the Brazilian population was higher, and most people didn't know, which would have lead to more hybrid immunity in that population. 

Line 268. I don't think the data supports this statement that there is significant waning against severe disease in the >65 yo group. Figure 2 does show some drop in percentage point VE, but not lower than 80% and with overlapping confidence intervals. The authors state on line 227: modest waning from 89.5% (95%CI 88.0 - 90.8) at 2-4 weeks to 85.1% (95%CI 83.5 - 86.6) at 13 weeks or more. This is only 4 percentage points drop. this statement should be modified lest it be misinterpreted as these vaccines are not working in this high risk age group. 

Line 293 and whole paragraph. Same comment about overstating of waning against severe disease in the elderly. 

Line 315. There is no evidence that VE differs by BA1 and BA2 - data from UK, Qatar. No date on other omicron lineages. Probably not the same amount of difference in VE as with different variants in circulation.

Line 333. This does not seem to be the case for this vaccine. You could delete this limitation.

Instead I think a significant limitation to mention is the limited amount of time of followup post booster, which limits the assessment of waning.

Table S3. The negative VE >20 weeks after primary series in Scotland should be discussed. This is not a plausible finding and suggests a bias among those first vaccinated. Moreover, this seems more of an issue among the older age groups, and also evident to a lesser degree after booster. Most other studies that initially showed a negative VE found that it was no longer negative after adding more omicron cases. Scotland is an outlier in not having the strongly negative VE disappear.

Table S4. Trend test for what? Not clear.

Reviewer #3: This is a valuable and important paper that addresses clinical efficacy of mRNA vaccine boosting after mRNA or ChAdOx1 primary series vaccination. The comparison of two countries, with broadly comparable results, gives confidence. 

Points

This is a comparison to unvaccinated subjects. The recommendation from the paper is that booster vaccination should be promoted. How do the findings compare to people who have primary series vaccination but no booster ?

Table 1 shows that the demographics of the case positive and use negative groups are somewhat dissimilar. Have these been assessed for statistical difference and could this influence the analysis.

Considerable emphasis is given to waning in subjects >65 years in Brazil whereas the values are not markedly dissimilar from the total cohort. Is there any reason why this was picked out ?

[LINK]

---

## [Decision Letter · Decision Letter 2]

21 Oct 2022

Dear Dr. Cerqueira-Silva,

Thank you very much for submitting your manuscript "Effectiveness of mRNA boosters after homologous primary series with BNT162b2 or ChAdOx1 against symptomatic infection and severe COVID-19 in Brazil and Scotland:  a test-negative design case-control study" (PMEDICINE-D-22-01640R2) for consideration at PLOS Medicine. 

[LINK]

In light of these reviews, I am afraid that we will not be able to accept the manuscript for publication in the journal in its current form, but we would like to consider a revised version that addresses the reviewers' and editors' comments. Obviously we cannot make any decision about publication until we have seen the revised manuscript and your response, and we plan to seek re-review by one or more of the reviewers. 

We expect to receive your revised manuscript by Nov 04 2022 11:59PM. Please email us (plosmedicine@plos.org) if you have any questions or concerns.

We look forward to receiving your revised manuscript. 

Sincerely,

Pippa

Philippa Dodd MBBS MRCP PhD 

PLOS Medicine

plosmedicine.org

GENERAL

Thank you for addressing previous reviewer and editor comments

Please address all editor and reviewer comments detailed below and in the attached file

DATA AVAILBILITY STATEMENT

Thank you for updating your data availability statement. Please also include this in your manuscript submission form – the older statement is still placed here – and remove from the end of the manuscript.

ABSTRACT

Please quantify main results with p-values as well as 95% CIs (this is currently a journal requirement)

When p<0.01 please report to 3 decimal places, and when p>/=.01 please report to 2 decimal places

Thank you for adding a limitation of your study – are there any others? Such as follow-up time post booster, inability to discriminate between omicron lineages etc. Please include all the primary limitations.

Where you report statistical results, please add a space between % symbols and CI i.e. line 55: “(95%CI…”. This is repeated throughout, please check and amend.

AUTHOR SUMMARY

Thank you for including an author summary. Please ensure it is structured according to PLOS Medicine’s style as described here: https://journals.plos.org/plosmedicine/s/revising-your-manuscript#loc-author-summary Specifically, please note the requirement for individually bulleted points and the requirement for the summary to be distinct from the abstract and accessible to the general reader.

METHODS and RESULTS

Please quantify main results with p-values as well as 95% CIs (this is currently a journal requirement)

Where p values are reported please also specify the statistical test used to determine it as well as the significance level used (eg p <0.05 two sided)

Line 140: “…for Brazil and Scotland” suggest “in” instead of “for” please revise

Line 177: “QCovid risk groups are…” I think it would be helpful to define the QCovid risk before you mention it. As written, it appears as though it is an after-thought and actually, it is vital for the reader to understand how the impact of co-morbidity in this group was accounted for. 

It might also be helpful to include the QCovid risk algorithm that you utilized in your analyses in the supporting files – I understand that you have made reference to it but, if possible, it would be helpful to the reader to include it. 

Line 206: “This study is reported following Strengthening….” Please place this statement at the beginning of the methods section

Line 209: “The original statistical plan is provided as supplementary appendix.” Suggest “prospective” instead of “original” – please revise throughout because as written, this statement implies that there is a revised analysis plan. Please move this sentence to the beginning of the section entitled “Statistical analysis” Please also indicate where in the appendix this information can be found i.e. S3 as per your manuscript.

In response to point 28 from the previous revision you state the following: “A: All data used in our analysis came from structured administrative databases. No detailed information about the definition of the characteristics was provided.” Please ensure that a relevant sentence is included in your methods section and indicate where you have placed it 

FIGURES & TABLES

Thank you for including appropriate captions for Figure 1 and Figure 2. In both, instead of “line” suggest “error bars represent 95% Wald confidence interval.” 

Please also include table captions, including for those in the supporting information, which clearly describe the content and include any abbreviations

Please define the numerical values, i.e. n (%), in table column headings (as you have [partially] done for table 1) as opposed to as a footnote – perhaps in the control column? 

Table 1 – please remove “N(%)” from the end of the table and include the (%) in the column headings

In all tables where you report 95% CIs please also report p-values – to conserve space and/or the need for splitting your tables, which are already very full with data, I would consider detailing the p-value significance level in the relevant caption and placing asterisks adjacent to the CIs to reflect the result. 

In the table captions please also report the statistical tests used to determine the p-value

Table S2: foot note reads “n (row percent%)” which is confusing to me, please revise for clarity and place this information in column headings (see above)

Table S4 – S8: all titles state the following “Vaccine Effectiveness % (95% CI) against…” suggest for clarity: “ Vaccine Effectiveness reported as percentages (95% CI) against…” or something similar

Figure 1, figure 2, figure S3B and figure S4: Please confirm that this shade of green is selected from a colour palate that this is accessible to those with color blindness – it appears to be a different shade from that used in figure S3A, at least on my monitor! 

Figure S2B: the test is very small and rather inaccessible to the reader, please revise

Please remove the data availability statement from the end of the manuscript and please remove the conflict-of-interest statement from the end of the manuscript. This information should be included in the manuscript submission form only and will be complied as metadata

REFERENCES

For in-text reference callouts please add a space between the text and the first of the parentheses. For example line 99: “…wane over time[4,5], prompting…” should read: “…wane over time [4,5], prompting…”. This error is repeated throughout, please check thoroughly and amend 

SOCIAL MEDIA

To help us extend the reach of your research, please provide any Twitter handle(s) that would be appropriate to tag, including your own, your coauthors’, your institution, funder, or lab. Please respond to this email with any handles you wish to be included when we tweet this paper.

Comments from the reviewers:

Reviewer #1: See attachment

Michael Dewey

[LINK]

---

## [Decision Letter · Decision Letter 3]

9 Dec 2022

Dear Dr. Cerqueira-Silva,

Thank you very much for re-submitting your manuscript "Effectiveness of mRNA boosters after homologous primary series with BNT162b2 or ChAdOx1 against symptomatic infection and severe COVID-19 in Brazil and Scotland:  a test-negative design case-control study" (PMEDICINE-D-22-01640R3) for review by PLOS Medicine.

I have discussed the paper with my colleagues and the academic editor and it was also seen again by the methodological reviewer. I am pleased to say that provided the remaining editorial and production issues are dealt with we are planning to accept the paper for publication in the journal.

[LINK]

We look forward to receiving the revised manuscript by Dec 16 2022 11:59PM.   

Sincerely,

Philippa Dodd, MBBS MRCP PhD

PLOS Medicine

plosmedicine.org

Requests from Editors:

GENERAL 

Thank you for your considerate and detailed responses to previous editor and reviewer requests which we believe have substantially improved your manuscript.

In parts the manuscript reads somewhat as though it has been written by two individuals. This is probably, and understandably, the case given the nature of the study design but it would be beneficial to improve the uniformity of the writing style.

Please respond to all comments detailed below, in full.

We note the reviewer comments regarding the use of two data sets. The editorial team has discussed this point and have split opinions. The academic editor notes a precedent for the use of two data sets in similar studies and the other reviewers think that the inclusion of the two data sets offer an advance, therefore we conclude that it perhaps amounts to a difference of opinion.

COMMENTS FROM THE ACADEMIC EDITOR

There is precedent for including multiple datasets in the same manuscript. I've seen it done to compare heterogeneous settings for HIV treatment outcomes. Here the vaccines administered, and SARS-CoV-2 transmission levels, are different across Scotland and Brazil so the comparisons can be useful.

CONFLICT OF INTEREST STATEMENT

Author Aziz Sheikh sits on the PLOS Medicine Editorial Board. Please add this statement to the manuscript's Competing Interests: "AS is an Academic Editor on PLOS Medicine's editorial board." I could not see that it had been listed.

ABSTRACT

We asked the following: “Thank you for adding a limitation of your study – are there any others? Such as follow up time post booster, inability to discriminate between omicron lineages etc. Please include all the primary limitations” but could not see any amendments, please revise accordingly.

Line 57 statistical reporting: “37.4% (95% CI 33.8 – 40.9; p<0.001)” thank you for including p-values and CIs suggest reporting as follows “37.4% (95% CI [33.8, 40.9], p<0.001). Please check and amend throughout the abstract, author summary and main manuscript to ensure consistency of reporting

AUTHOR SUMMARY

Line 82: as above, “(from 51.6% [95% confidence interval:CI - 51.0 – 52.2] to 4.2% [95% CI 0.7 – 7.6])” your statistical reporting is rather difficult to interpret, and the use of multiple hyphens can lead to confusion between negative values and intervals. 

Suggest the following: “from 51.6%, 95% confidence interval (CI): [51.0, 52.2], to 4.2% [95% CI: 0.7, 7.6])” . Note the absence of the first set of parentheses. Please check and revise throughout 

Please include p-values where you report 95% CIs

Line 84: sentence beginning “In these periods…” suggest make a separate bulleted for this statement

Line 93: “Similar findings in two very distinct countries allow us to draw reliable results because each country has different key sources of potential bias unrelated to each other” 

It would be helpful to clarify “potential bias unrelated to each other”, this statement is vague and unclear to the general reader. In addition, formal comparisons between the two data sets were not made so this statement can’t be fully substantiated by the dataset presented. 

I would perhaps reword this statement and consider removing it from the conclusions to the previous section “what did the researchers do and find” as it seems partly aimed at justifying the use of the two data sets.

METHODS and RESULTS

Please amend statistical reporting as described above

TABLES

Table S2: please define numerical values within and outside of parentheses – suggest “n (%)” as in other tables

FIGURES

Figure S2 B) – please define TND and RT PCR in an appropriate caption

SOCIAL MEDIA

To help us extend the reach of your research, please provide any Twitter handle(s) that would be appropriate to tag, including your own, your coauthors’, your institution, funder, or lab. Please respond to this email with any handles you wish to be included when we tweet this paper.

Comments from Reviewers:

Reviewer #1: The authors have addressed my points. I still do not see what the added value of having two cohorts presented together is but I seem to be alone in my confusion.

Michael Dewey

[LINK]

---

## [Editor Report · Decision Letter 4]

13 Dec 2022

Dear Dr Cerqueira-Silva, 

On behalf of my colleagues and the Academic Editor, Dr. Amitabh Suthar, I am pleased to inform you that we have agreed to publish your manuscript "Effectiveness of mRNA boosters after homologous primary series with BNT162b2 or ChAdOx1 against symptomatic infection and severe COVID-19 in Brazil and Scotland:  a test-negative design case-control study" (PMEDICINE-D-22-01640R4) in PLOS Medicine.

Please note this final amendment in the abstract methods and findings section at line 10:

 “…booster was 51.6%, 95% confidence interval (CI): ([ 51.0, 52.2], p<0.001) in Brazil…” should read: “booster was 51.6%, (95% confidence interval (CI): [51.0, 52.2], p<0.001) in Brazil” note the alteration to the placement of circular parentheses. Please amend prior to publication.

PRESS

Sincerely, 

Philippa Dodd, MBBS MRCP PhD 

PLOS Medicine